# The Effect of Anti-Tumor Necrosis Factor-Alpha Therapy within 12 Weeks Prior to Surgery on Postoperative Complications in Inflammatory Bowel Disease: A Systematic Review and Meta-Analysis

**DOI:** 10.3390/jcm11236884

**Published:** 2022-11-22

**Authors:** Kamacay Cira, Marie-Christin Weber, Dirk Wilhelm, Helmut Friess, Stefan Reischl, Philipp-Alexander Neumann

**Affiliations:** 1Department of Surgery, School of Medicine, Klinikum Rechts der Isar, Technical University of Munich, 81675 Munich, Germany; 2Institute of Diagnostic and Interventional Radiology, School of Medicine, Klinikum Rechts der Isar, Technical University of Munich, 81675 Munich, Germany

**Keywords:** inflammatory bowel disease, Crohn’s disease, ulcerative colitis, anti-tumor necrosis factor-alpha, colorectal surgery, postoperative complications

## Abstract

The rate of abdominal surgical interventions and associated postoperative complications in inflammatory bowel disease (IBD) patients is still substantially high. There is an ongoing debate as to whether or not patients who undergo treatment with anti-tumor necrosis factor-alpha (TNF-α) agents may have an increased risk for general and surgical postoperative complications. Therefore, a systematic review and meta-analysis was conducted in order to assess the effect of anti-TNF-α treatment within 12 weeks (washout period) prior to abdominal surgery on 30-day postoperative complications in patients with IBD. The results of previously published meta-analyses examining the effect of preoperative anti-TNF-α treatment on postoperative complications reported conflicting findings which is why we specifically focus on the effect of anti-TNF-α treatment within 12 weeks prior to surgery. PubMed, Cochrane, Scopus, Web of Science, World Health Organization Trial Registry, ClinicalTrials.gov and reference lists were searched (June 1995–February 2022) to identify studies, investigating effects of anti-TNF-α treatment prior to abdominal surgery on postoperative complications in IBD patients. Pooled odds ratios (OR) with 95% confidence intervals (CI) were calculated and subgroup analyses were performed. In this case, 55 cohort studies (22,714 patients) were included. Overall, postoperative complications (OR, 1.23; 95% CI, 1.04–1.45; *p* = 0.02), readmission (OR, 1.39; 95% CI, 1.11–1.73; *p* = 0.004), and intra-abdominal septic complications (OR, 1.89; 95% CI, 1.44–2.49; *p* < 0.00001) were significantly higher for anti-TNF-α-treated patients. Significantly higher intra-abdominal abscesses and readmission were found for anti-TNF-α-treated CD patients (*p* = 0.05; *p* = 0.002). Concomitant treatment with immunosuppressives in <50% of anti-TNF-α-treated patients was associated with significantly lower mortality rates (OR, 0.32; 95% CI, 0.12–0.83; *p* = 0.02). Anti-TNF-α treatment within 12 weeks prior to surgery is associated with higher short-term postoperative complication rates (general and surgical) for patients with IBD, especially CD.

## 1. Introduction

With worldwide increasing incidence and prevalence, inflammatory bowel diseases (IBD), encompassing Crohn’s Disease (CD) and Ulcerative Colitis (UC), belong to a group of diseases with substantial social and economic burden on health care systems and governments. [1] Compared to the 1990s, the life-time risk of undergoing a major abdominal surgery for IBD has decreased in the last decade. [2] Consequently, the treatment with biological drugs has continuously increased over these years and has become an established practice in the treatment of IBD nowadays. [2] Tumor necrosis factor-alpha (TNF-α) inhibitors were among the first FDA approved biologic drugs in the treatment for IBD. Today, up to 40% of CD patients and up to 16% of UC patients are under treatment with anti-TNF-α biologics such as Adalimumab, Infliximab, Golimumab, Certolizumab (pegol) or one of its biosimilars. However, in comparison with the general population, the 5-year surgery rates with its accompanying potential postoperative complications remain substantial for IBD patients. [2] Previously conducted meta-analyses, analyzing the effect of preoperative anti-TNF-α treatment on postoperative complications has reported conflicting findings [3,4,5,6,7,8,9,10,11,12,13,14,15,16,17].

The latter is mostly the result of including different preoperative drug withdrawal periods in the analyses. [3,4,5,6,7,8,9,10,11,12,13,14,15,16,17] In order to find consistent values regarding the effect of preoperative anti-TNF-α treatment on postoperative complications and to ensure better comparisons of data, we chose the objectifiable anti-TNF-α drug washout period of 12 weeks [18] as the preoperative cut-off value in this analysis. 

Therefore, the aim of this study was to evaluate the effect of anti-TNF-α treatment exclusively within the washout period of 12 weeks (3 months) [18] prior to abdominal surgery, on a 30-day general and surgical postoperative complication rate in IBD patients. A systematic review and meta-analysis were conducted, comparing summary effect sizes, calculating the pooled odds ratio (OR) with 95% confidence intervals (CI) and performing subgroup analyses with subgroups stratified by IBD subtype, surgical approach (open versus (vs.) laparoscopic), elective vs. emergency surgery, protective ileostomy use and concomitant use of corticosteroids and/or immunomodulatory drugs.

## 2. Methods

This systematic review and meta-analysis were conducted and reported according to the recommendations in the Cochrane Handbook for Reviews of Interventions [19] and the Preferred Reporting Items for Systematic review and Meta-Analyses (PRISMA) statement 2020 [20].

### 2.1. Eligibility Criteria

All observational studies (prospective or retrospective comparative cohort or case-control studies), nested case-control studies, non-randomized controlled trials, randomized controlled trials and cross-sectional studies were included based on the following criteria: examined humans (of which the majority were ≥18 years old), being published in English or German language, being available as full-text article in the electronic medical databases between 1 June 1995 and 17 February 2022, patients undergoing any intestinal surgical procedures for IBD (CD/UC/Indeterminate Colitis (IC)), an intervention group including patients who either received anti-TNF-α biologics within the washout period of 12 weeks (3 months) [18] prior to surgery or had a detectable serum concentrations (≥0.98 µg/mL) [21] of these drugs at surgery (regardless of anti-TNF-α biologic preparation and dose), a control group including patients who either did not received any biologic therapy before surgery or had no detectable serum concentrations of an anti-TNF-α biologic (<0.98 µg/mL) [21] at surgery, and a reported exactly 30-day postoperative complication rate. If studies reported two or more discrete data sets, these sets were separately included for the analysis.

The exclusion criteria comprised studies without a control group, abstracts and conference proceedings, reviews and meta-analyses, case-reports and case-series, in-vivo, ex-vivo and in-vitro studies, animal studies, predominantly pediatric patients being studied, concomitant use of any biologic drug other than anti-TNF-α biologics, the intervention groups’ last exposure to anti-TNF-α biologics more than 12 weeks (3 months) [18] prior to intestinal surgery, and a follow-up period below or above 30 days after surgery for both the intervention and control groups.

### 2.2. Search Strategy

A systematic literature search was performed for studies published in the electronic medical databases PubMed (MEDLINE), Web of Science, Cochrane Library and Cochrane central register of controlled trials, Scopus, ClinicalTrials.gov and the World Health Organization Trials Registry, using predefined search items for each database (Appendix A). The reference lists of the included studies were examined manually, and an additional web search was conducted to ensure that potentially relevant studies were not missed (Appendix A) [22,23,24,25,26,27,28,29,30,31,32,33,34,35,36,37,38,39,40,41,42]. In the case of insufficient or inadequate data presentation, authors were contacted to provide the required information. The final search was conducted on 17 February 2022.

### 2.3. Selection Process

All studies were assessed manually and independently by two investigators (surgical resident: K.C., radiology resident: S.R.) and were exported to the reference management tool EndNote X9 (EndNote X9; The EndNote Team, Clarivate 2013; Philadelphia, PA, USA). According to this studies’ predefined eligibility criteria, titles and abstracts were assessed, excluding both duplicates and studies not coinciding with the eligibility criteria. Finally, the remaining full-text articles were retrieved and evaluated for eligibility. The disagreements concerning eligibility were resolved in consensus with a third investigator (surgical specialist: P-A.N.), who independently assessed the accuracy of search results.

### 2.4. Data Collection Process

The data were collected and analyzed independently by two investigators (K.C., S.R.) onto a Microsoft Excel spreadsheet (Home and Student 2019 edition; Microsoft, Redmond, WA) and was assessed independently for accuracy by a third investigator (P-A.N.). The emerging discrepancies in relation to data extracted were discussed and resolved by consensus of all three investigators. 

### 2.5. Data Items

The following data were collected for each study if available: author, year, and country of publication, study design and inclusion period, medical treatment and time frame in which drugs were used preoperatively, number of patients in the intervention and control group, patients’ baseline and surgical characteristics, and postoperative follow-up period and postoperative complication rates.

In order to capture all complications while ensuring comparability of data, postoperative complication rates were analyzed according to the definitions used in the included studies.

For this review, 30-day postoperative complication rates were quantified and analyzed, including: (1)general overall complication rates:
overall postoperative complicationsoverall infectious postoperative complicationsoverall Clavien-Dindo minor and major complicationsreadmission rates, reoperation rates, mortality rates
(2)surgical-site complication rates:
overall infectious surgical complicationsintra-abdominal septic complications, anastomotic leakages (AL), intra-abdominal abscesses (without drainage)surgical-site infections (SSI) (incisional- and deep or organ space)postoperative hemorrhages, ileus, small bowel obstructions, fistula formations
(3)non-surgical-site complication rates:
overall infectious and non-infectious non-surgical-site complicationsthrombosis, cardiovascular complicationspneumonia, urinary tract infections and sepsis

In order to serve as an orientation for surgeons and provide an implication into surgical practice, this review and the resulting meta-analysis primarily focused on 30-day postoperative general and surgical complication rates.

### 2.6. Assessment of Risk of Bias and Quality of Included Studies

The qualities of the included studies were assessed independently by two investigators (K.C., S.R.) using the Newcastle-Ottawa scale (NOS) for cohort studies [43], a valid and commonly used score applied for observational studies. [4,5,6,7,9,16,17,43] The investigators’ (K.C., S.R.) judgements were conclusively justified (Appendix A) and any disagreements were resolved in consensus with a third investigator (P-A.N.). A NOS score of >7 was defined as high-quality, 5–7 as moderate-quality and <5 as low-quality study. 

Potential risk factors for postoperative complications within each study group were defined a priori as: tobacco use in >50% of patients, open surgical approach and/or conversion to open surgery in >50% of patients (OA), elective and emergency surgery (ELEMS) vs. exclusively elective surgery (EL), performance of a temporary protective ileostomy in < 50% of patients or no use at all (PRI), Body-Mass-Index (BMI) of >25 kg/m^2^ or <18.5 kg/m^2^, dysplasia or malignancy in >50% of patients; perforating or penetrating disease as indication for surgery in >50% of patients, concomitant corticosteroid and/or immunomodulatory agent use in >50% of patient (CSIM), laboratory values (median or mean C-reactive protein concentration of >10 mg/L, white blood cell count > 11 × 10^9^ cells/L or <4 × 10^9^ cells/L, hemoglobin value of <10 g/dL, and albumin value of <3 g/dL and platelet count of <150 × 10^3^/µL).

### 2.7. Synthesis Methods

The heterogeneity between studies was analyzed using the statistical *I*^2^ test [44], considering a *I*^2^ of ≥50% as substantial heterogeneity. [44] In case of significant or substantial heterogeneity among the included studies, sensitivity analysis was conducted by evaluating the effect of excluding one study at a time on the pooled OR. According to the recommendations of the Cochrane Handbook for Reviews of Interventions [19], potential publication biases were assessed for outcomes reported by ≥10 studies by applying the Egger’s test [45] for funnel plot asymmetry. 

The prespecified subgroup analyses were performed for outcomes reported by ≥ 5 studies to evaluate the influence of IBD subtypes and potential risk factors on the 30-day postoperative complications. The differences in the outcomes between these subgroups were assessed and reported using the test for subgroup differences (TSD). For subgroups stratified by IBD subtypes, sensitivity analysis was conducted by excluding IBD mixed populations, for which data of the separate IBD subtypes could not be retrieved. Further sensitivity analysis was conducted for subgroups stratified by prespecified risk factors by excluding studies, for which data of studied risk factor were not available. 

The results with a *p*-value of <0.05 were considered significant. All statistical analyses were carried out using the Review Manager software version 5.3 (Nordic Cochrane Centre, Copenhagen, Denmark) and JASP Team (2021; JASP (Version 0.16) [Computer software]). In the case of significant or substantial heterogeneity (*I*^2^ ≥ 50%) of included studies, random-effects model (RE) was used to conduct the meta-analysis. With a non-significant *I*^2^ < 50% of included studies, fixed-effects model (FE) was used to conduct the meta-analysis. Using either RE or FE meta-analysis, pooled OR with 95% CI were summarized and depicted in a forest plot.

## 3. Results

### 3.1. Study Selection

As depicted in Figure 1, 914 studies were identified through electronic medical database search and another 21 studies through manual search (other methods). After removing 181 duplicates, titles and abstracts of 733 articles were screened manually of which 590 were excluded. For the full-text review 137 out of 143 studies (electronic medical database search) and 9 out of 21 studies (other methods) could be retrieved. 

Finally, 51 cohort studies [21,22,23,46,47,48,49,50,51,52,53,54,55,56,57,58,59,60,61,62,63,64,65,66,67,68,69,70,71,72,73,74,75,76,77,78,79,80,81,82,83,84,85,86,87,88,89,90,91,92,93] fulfilled the inclusion criteria, three [21,55,89] of which included at least two discrete data sets and were subjected to qualitative and quantitative analysis (Figure 1).

### 3.2. Study Characteristics

The systematic review and meta-analysis investigate 47 retrospective [21,22,23,46,47,48,50,51,53,54,55,56,57,58,60,61,62,63,64,65,66,68,69,70,71,72,73,74,75,76,77,78,79,80,81,82,83,84,85,86,87,88,89,90,91,92,93] and four prospective [49,52,59,67] studies. 

Overall, 22,714 patients were included in the 51 studies (54 discrete data sets) [21,22,23,46,47,48,49,50,51,52,53,54,55,56,57,58,59,60,61,62,63,64,65,66,67,68,69,70,71,72,73,74,75,76,77,78,79,80,81,82,83,84,85,86,87,88,89,90,91,92,93], 4417 patients were treated with an anti-TNF-α agent within 12 weeks prior to abdominal surgery (intervention group) and 18,297 patients did not receive anti-TNF-α treatment (control group). TNF-α inhibitors that were investigated included: Adalimumab, Certolizumab (pegol), Golimumab, Infliximab and/or its biosimilars [21,22,23,46,47,48,49,50,51,52,53,54,55,56,57,58,59,60,61,62,63,64,65,66,67,68,69,70,71,72,73,74,75,76,77,78,79,80,81,82,83,84,85,86,87,88,89,90,91,92,93].

The most commonly studied IBD subtype was CD, including 28 studies (30 discrete data sets) [21,22,23,46,47,49,50,51,56,57,58,60,61,64,65,66,68,69,70,73,77,79,80,81,84,88,89,90,91,92] investigating 2272 patients in the intervention and 8039 patients in the control group. Another 13 studies (16 discrete data sets) [21,48,53,54,55,59,71,72,76,82,83,85,86,89,93] investigated 1600 UC (and IC) patients in the intervention and 8734 UC (and IC) patients in the control group. Ten studies [21,52,62,63,67,74,75,78,87,89] examined a mixed IBD patient group, out of which eight studies [52,62,63,67,74,75,78,87] did not further specify the underlying disease (545 patients in the intervention and 1524 in the control group) (Table 1).

All included studies analyzed the short-term postoperative complication and 30-day complication rates (Table 2 and Appendix A).

### 3.3. Results of Synthesis

#### 3.3.1. Analysis of General Postoperative Complications

##### Overall Postoperative Complications

Overall, 27 studies (30 discrete data sets) [21,22,23,47,48,49,51,52,54,55,56,57,58,59,61,62,63,67,69,70,74,75,81,86,89,90,91] reported 30-day overall postoperative complications occurring in 678 of 2588 (26.2%) patients in the intervention group and 2059 of 11,004 (18.7%) in the control group. OPC were significantly higher for patients preoperatively treated with anti-TNF-α agents, using RE meta-analysis (OR, 1.23; 95% CI, 1.04–1.45; *p* = 0.02) (Figure 2). The studies showed a significant heterogeneity (*I*^2^ = 44%; *p* = 0.006) but sensitivity analysis showed a significant reduction in heterogeneity with the exclusion of the study by Brouquet et al., 2018 [49] (OR, 1.17; 95% CI, 1.01–1.36; *p* = 0.04; *I*^2^ = 28%; *p* = 0.08). No publication bias was observed (Egger’s test: *p* = 0.204). 

The subgroup analyses found no subgroup difference for subgroups stratified by IBD subtypes, OA, ELEMS, PRI and CSIM (Appendix A).

##### Overall Postoperative Infectious Complication

In this case, 20 studies (22 discrete data sets) [21,23,48,53,54,58,59,62,63,64,69,71,74,75,76,79,80,86,89,91] reported 30-day overall postoperative infectious complications in 302 of 2208 (13.7%) patients in the intervention group and 858 of 9333 (9.2%) patients in the control group. No significant differences were found between the examined group, using RE meta-analysis (OR, 1.16; 95% CI, 0.92–1.45; *p* = 0.21) (Appendix A).

The studies showed a significant heterogeneity (*I*^2^ = 43%; *p* = 0.02) which remained stable throughout sensitivity analysis. No publication bias was observed (Egger’s test: *p* = 0.154) and subgroup analyses found no subgroup difference for subgroups stratified by IBD subtypes, OA, ELEMS, PRI and CSIM (Appendix A).

##### Overall Clavien-Dindo Major Complications

A total of 7 studies [23,60,62,66,69,79,81] reported 30-day overall postoperative Clavien-Dindo major complications, manifesting in 101 of 564 (17.9%) patients in the anti-TNF-α group and 176 of 1243 (14.1%) patients in the control group. No difference in OC-DMC was found between the studied group, using FE meta-analysis (OR, 1.13; 95% CI, 0.85–1.50; *p* = 0.40). 

The studies were homogeneous (*I*^2^ = 0%; *p* = 0.91). (Appendix A) subgroup analyses found no subgroup difference for subgroups stratified by IBD subtypes; OA; ELEMS; PRI and CSIM (Appendix A).

##### Readmission

In this case, 30-day postoperative readmission rates were reported by 15 studies (17 data sets) [21,46,52,55,59,60,61,62,71,75,78,79,87,88,89], with 162 of 1192 (13.6%) patients in the intervention and 259 of 2580 (10.03%) patients in the control group being readmitted. Readmission was presented to be significantly higher for patients in the intervention group, using FE meta-analysis (OR, 1.39; 95% CI, 1.11–1.73; *p* = 0.004) (Figure 3). The studies were homogeneous (*I*^2^ = 0%; *p* = 0.45) and no publication bias was observed (Egger’s test: *p* = 0.994).

After conducting sensitivity analysis, the subgroup analyses found significant subgroup differences for subgroups stratified by IBD subtypes (TSD: *p* = 0.15; sensitivity analysis: *p* = 0.05), OA (TSD: *p* = 0.06; sensitivity analysis: *p* = 0.02) and PRI (TSD: *p* = 0.06; sensitivity analysis: *p* = 0.03). A significantly higher readmission rate was found for patients in the intervention group if the underlying disease was CD (OR, 1.77; 95% CI, 1.23–2.52; *p* = 0.002), if OA was conducted in >50% of patients (OR, 2.01; 95% CI, 1.32–3.08; *p* = 0.001) and if PRI was used in <50% (OR, 1.79; 95% CI, 1.32–2.42; *p* = 0.0002) (Table 3).

##### Reoperation

Overall, 14 studies (15 data sets) [46,50,55,60,61,62,69,72,73,74,75,79,87,89] reported 30-day postoperative reoperation rates in 142 of 1493 (9.5%) patients in the intervention and 534 of 5400 (9.9%) patients in the control group. No significant differences were found between the studied groups, using FE meta-analysis (OR, 1.02; 95% CI, 0.82–1.26; *p* = 0.88) (Appendix A). The studies were homogeneous (*I*^2^ = 0%; *p* = 0.80) and no significant publication bias was observed (Egger’s test: *p* = 0.439). 

The subgroup analyses found no subgroup difference for subgroups stratified by IBD subtypes, OA, ELEMS, PRI and CSIM (Appendix A).

##### Mortality

In this case, 30-day postoperative mortality was reported by 13 studies [21,46,55,58,60,61,62,63,70,72,73,79,87], occurring in 13 of 1432 (0.9%) patients in the anti-TNF-α group and 95 of 5538 (1.7%) patients in the control group. No significant difference was found between the studied groups, using FE meta-analysis (OR, 0.73; 95% CI, 0.41–1.30; *p* = 0.28) (Appendix A). The studies were homogeneous (*I*^2^ = 30%; *p* = 0.14) and no significant publication bias was observed (Egger’s test: *p* = 0.361). 

The subgroup analyses found a significant subgroup difference for subgroups stratified by CSIM (TSD: *p* = 0.02; sensitivity analysis: *p* = 0.006). Mortality was significantly lower for patients in the intervention group if concomitant corticosteroids or immunomodulatory drugs were used in just a minority of patients or not at all (OR, 0.32; 95% CI, 0.12–0.83; *p* = 0.02). No sensitivity analysis stable subgroup differences were found for subgroups stratified by IBD subtypes, OA, ELEMS and PRI (Appendix A).

#### 3.3.2. Analysis of Surgical Postoperative Complications

##### Overall Infectious Surgical-Site Complications 

A total of 11 studies [23,53,60,61,65,71,77,82,83,84,85] reported 30-day overall postoperative infectious surgical-site complications, appearing in 112 of 625 (19.9%) patients in the intervention and 324 of 1630 (19.8%) in the control group. No significant differences were found between anti-TNF-α and control group, using RE meta-analysis (OR, 0.81; 95% CI, 0.45–1.45; *p* = 0.48) (Appendix A).

The studies showed a significant substantial heterogeneity (*I*^2^ = 76%; *p* < 0.0001), still results remained stable throughout sensitivity analysis. No significant publication bias was observed (Egger’s test: *p* = 0.439).

The subgroup analyses found a significant subgroup difference for subgroups stratified by ELEMS (TSD: *p* < 0.00001; sensitivity analysis: *p* < 0.00001). Anti-TNF-α treated CD patients who had undergone elective surgery presented with significantly higher OISSC (OR, 2.02; 95% CI, 1.31–3.11; *p* = 0.002), while CD and UC patients who underwent both elective and emergency surgery developed OISSC significantly less (OR, 0.31; 95% CI, 0.18–0.55; *p* < 0.0001). No significant sensitivity analysis stable subgroup differences were found for subgroups stratified by IBD subtypes, OA, PRI and CSIM (Appendix A).

##### Intra-Abdominal Septic Complications

In this case, 30-day postoperative intra-abdominal septic complications were reported by 9 studies [49,52,64,68,70,79,80,90,92], manifesting in 108 of 830 (13.01%) patients in the intervention and 164 of 2040 (8.04%) patients in the control group. Using FE meta-analysis, patients in the anti-TNF-α group had significantly higher postoperative intra-abdominal septic complications (OR, 1.89; 95% CI, 1.44–2.49; *p* < 0.00001). The studies were homogeneous (*I*^2^ = 47%; *p* = 0.06) (Figure 4).

No subgroup difference was found in subgroup analysis for subgroups stratified by IBD subtypes (TSD: *p* = 0.47), still 8 [49,64,68,70,79,80,90,92] out of 9 studies [49,52,64,68,70,79,80,90,92] examined CD patient groups. Excluding the only IBD mixed patient group (study by El-Hussuna et al., 2018 [52]) resulted in a significantly higher postoperative intra-abdominal septic complication rate for anti-TNF-α treated CD patients in comparison to the control group (OR, 1.92; 95% CI, 1.45–2.53; *p* < 0.00001). No sensitivity analysis stable subgroup differences were found for subgroups stratified by OA, ELEMS, PRI and CSIM (Appendix A).

##### AL or Abscess

A total of 21 studies (22 data sets) [23,46,50,53,55,60,61,62,63,69,72,73,74,75,78,79,80,87,89,90,93] reported 30-day postoperative AL or abscesses in 93 of 1827 (5.1%) patients in the intervention group and 313 of 7078 (4.4%) patients in the control group. No significant differences were found between the studied groups, using FE model meta-analysis (OR, 1.02; 95% CI, 0.79–1.32; *p* = 0.86) (Appendix A). Studies were homogeneous (*I*^2^ = 0%; *p* = 0.88) and no significant publication bias was observed (Egger’s test: *p* = 0.216). 

The subgroup analysis showed no subgroup differences for subgroups stratified by the IBD subtype, ELEMS and CSIM (Appendix A).

##### Intra-Adominal Abscesses

Overall, 18 studies (19 data sets) [23,46,48,50,53,54,55,59,60,61,62,71,74,78,79,80,89,93] reported the presence of intra-abdominal abscesses 30-days postoperatively, occurring in 81 of 1188 (6.8%) patients in the anti-TNF-α group and 229 of 3333 (6.9%) patients in the control group. No significant difference was found between the two groups, using FE model meta-analysis (OR, 1.19; 95% CI, 0.89–1.60; *p* = 0.24) (Appendix A). The studies were homogeneous (*I*^2^ = 0%; *p* = 0.76) and no significant publication bias was observed (Egger’s test: *p* = 0.85). 

After conducting sensitivity analysis, subgroup analysis showed a significant subgroup difference for subgroups stratified by IBD subtypes (TSD: *p* = 0.15; sensitivity analysis: *p* = 0.05) and PRI (TSD: *p* = 0.13; sensitivity analysis: *p* = 0.05). Intra-abdominal abscess rate was significantly higher for anti-TNF-α treated patients, if the underlying disease was CD (OR, 1.50; 95% CI, 1.0–2.25; *p* = 0.05) and if only a minority of patients received a protective ileostomy (OR, 1.58; 95% CI, 1.06–2.37; *p* = 0.02). 

No subgroup differences were found for subgroups stratified by OA, ELEMS, and CSIM (Table 3).

#### 3.3.3. Analysis of Other Postoperative Complications

No significant differences were found between the intervention and control group for: (1)Overall non-infectious postoperative complications [48,54,71,89] (FE: OR, 0.85; 95% CI, 0.56–1.29; *p* = 0.45; *I*^2^ = 23%; *p* = 0.27) (Appendix A).(2)Overall Clavien-Dindo minor complications [23,60,62,66,69] (FE: OR, 1.30; 95% CI, 0.98–1.72; *p* = 0.06; *I*^2^ = 34%; *p* = 0.19) (Appendix A).(3)Superficial SSI [23,52,65,71,74,82,83,84,85,89] (RE: OR, 0.67; 95% CI, 0.32–1.40; *p* = 0.29; *I*^2^ = 69%; *p* = 0.0004–stable throughout sensitivity analysis; Egger’s Test: *p* = 0.643; no subgroup difference in subgroup analysis for subgroups stratified by IBD subtype: TSD: *p* = 0.93; sensitivity analysis: *p* = 0.74) (Appendix A).(4)Deep or organ space SSI [23,82,83,84,85] (FE: OR, 0.76; 95% CI, 0.48–1.22; *p* = 0.26; *I*^2^ = 0%; *p* = 0.54) (Appendix A).(5)Fistula formation [74,79,80] (FE: OR, 0.65; 95% CI, 0.17–2.47; *p* = 0.52; *I*^2^ = 0%; *p* = 0.83) (Appendix A).(6)Ileus [23,48,55,59,62,66,70,74,75,89] (FE: OR, 1.11; 95% CI, 0.81–1.51; *p* = 0.52; *I*^2^ = 0%; *p* = 0.15; Egger’s Test: *p* = 0.115; no subgroup difference in subgroup analysis for subgroups stratified by IBD subtype: TSD: *p* = 0.76; sensitivity analysis: *p* = 0.5) (Appendix A).(7)Small bowel obstruction [23,55,60,61,62,74] (FE: OR, 0.76; 95% CI, 0.43–1.32; *p* = 0.33; *I*^2^ = 0%; *p* = 0.86; no subgroup difference in subgroup analysis for subgroups stratified by IBD subtype: TSD: *p* = 0.69; sensitivity analysis: *p* = 0.41) (Appendix A).(8)Postoperative hemorrhage [23,55,63,75,79,89] (FE: OR, 0.83; 95% CI, 0.49–1.41; *p* = 0.50; *I*^2^ = 0%; *p* = 0.86; no subgroup difference in subgroup analysis for subgroups stratified by IBD subtype: TSD: *p* = 0.37; sensitivity analysis: *p* = 1.0) (Appendix A).(9)Overall infectious non-surgical-site complications [53,71,78] FE: OR, 1.14; 95% CI, 0.43–3.03; *p* = 0.79; *I*^2^ = 0%; *p* = 0.53) (Appendix A).(10)Thrombosis [48,62,63,71,75,79,93] (FE: OR, 1.30; 95% CI, 0.75–2.25; *p* = 0.34; *I*^2^ = 14%; *p* = 0.34; no subgroup difference in subgroup analysis for subgroups stratified by IBD subtype: TSD: *p* = 0.24; sensitivity analysis: *p* = 0.14) (Appendix A).(11)Cardiovascular complications [62,63,70,79] (FE: OR, 0.62; 95% CI, 0.23–1.73; *p* = 0.36; *I*^2^ = 0%; *p* = 0.96) (Appendix A).(12)Pneumonia [48,50,55,59,60,61,62,79,87,89] (FE: OR, 0.83; 95% CI, 0.48–1.46; *p* = 0.52; *I*^2^ = 0%; *p* = 0.91; no subgroup difference in subgroup analysis for subgroups stratified by IBD subtype: TSD: *p* = 0.85; sensitivity analysis: *p* = 0.58) (Appendix A).(13)Urinary tract infection [46,48,53,55,60,61,62,87,89] (FE: OR, 1.02; 95% CI, 0.54–1.91; *p* = 0.96; *I*^2^ = 0%; *p* = 0.65; no subgroup difference in subgroup analysis for subgroups stratified by IBD subtype: TSD: *p* = 0.68; sensitivity analysis: *p* = 0.76) (Appendix A).(14)Sepsis [23,46,48,53,74,79] (FE: OR, 1.24; 95% CI, 0.75–2.04; *p* = 0.41; *I*^2^ = 0%; *p* = 0.44; no subgroup difference in subgroup analysis for subgroups stratified by IBD subtype: TSD: *p* = 0.90; sensitivity analysis: *p* = 0.67) (Appendix A).

## 4. Discussion

This systematic review and meta-analysis give an extensive overview on the 30-day postoperative complication rates in IBD patients following preoperative anti-TNF-α treatment within the washout period of 12 weeks [18] prior to abdominal surgery.

The results of previously published meta-analyses reported conflicting findings. Some studies reported a significant increase in overall postoperative complications [7,8,9,16] or overall infectious postoperative complications [3,6,7,9,16] for anti-TNF-α treated patients, others did not. [3,5,12,15,17]. This meta-analysis found significantly higher overall postoperative complications, readmission rates and intra-abdominal septic complications for anti-TNF-α treated patients (Figure 2, Figure 3 and Figure 4; Table 3; Appendix A). The analysis for overall postoperative complications showed a significant substantial heterogeneity which was significantly reduced when removing the study by Brouquet et al., 2018 [49] (Figure 2; Appendix A).

In this specific study, the authors reported an increased risk for overall morbidity and intra-abdominal septic complications for anti-TNF-α treated CD patients after ileocolic resection, regardless of disease severity. Since the studies’ authors did not specify overall postoperative morbidity further, this study might have introduced the observed heterogeneity [49].

In contrast, no significant differences were found between the studied groups for overall infectious complications, Clavien-Dindo major complications, reoperation rates and AL, even after performing thorough subgroup analysis stratified by disease subtypes and potential risk factors (Appendix A).

The significantly higher postoperative intra-abdominal abscesses and readmission rates were found in the subgroup analyses for patients in the intervention group, when the underlying disease was CD or when only a minority of patients received a protective ileostomy after surgery. In addition, an open surgical approach in >50% of patients were found to be associated with significantly higher readmission rates for patients in the intervention group (Figure 3; Table 3; Appendix A). The major surgical procedure for the anti-TNF-α treated group with low protective ileostomy formation (<50% of patients) and significantly higher intra-abdominal abscess rates were ileocolic resections [46,50,60,61,78,79,80] and colectomies [54] with primary anastomosis, mostly conducted in CD patients [46,50,60,61,79,80] (Table 3).

Interestingly, Myrelid et al., 2012 [94] investigated 132 CD patients who underwent ileocecal or ileocolonic resection for CD and found the 30-day postoperative anastomotic complications and reoperation rate to be significantly lower for patients undergoing high-risk resections with split-stoma and delayed anastomosis compared to those with primary anastomosis. Furthermore, the authors reported the risk of anastomotic complications to increase with the number of identified preoperative risk factors [94].

Even though this meta-analysis found no association between preoperative anti-TNF-α treatment and AL (Appendix A), significantly increased intra-abdominal abscesses were found in the anti-TNF-α treated CD group with low protective ileostomy formation (<50% of patients). The benefit of a temporary ileostomy for patients with CD has not been conclusively clarified and is overall rarely used in everyday surgical practice. 

In fact, previously conducted meta-analyses [6,7] had reported significantly higher intra-abdominal abscess rates, but significantly increased readmission rates have not been related to preoperative anti-TNF-α therapy previously [15].

Likewise, none of the previously conducted meta-analyses [11,15] reported significant differences in postoperative mortality between the examined groups, which corresponds to the primary finding in this study. However, subgroup analysis found significantly lower mortality rates for the anti-TNF-α-treated patient group when <50% of them were treated concomitantly with corticosteroids and/or immunomodulatory drugs (Appendix A).

This result suggests that the concomitant perioperative immunosuppression in patients preoperatively treated with anti-TNF-α biologics could be detrimental and should therefore be kept to a minimum, in order to increase the chance of overall postoperative survival. On the other hand, a certain protective effect of perioperative immunosuppression can be presumed for IBD patients not treated with anti-TNF-α biologics preoperatively.

No other general (Appendix A), surgical (Appendix A) or non-surgical complications (Appendix A) were found to be associated with preoperative anti-TNF-α therapy.

The results of these studies’ analyses have certain limitations that need to be addressed. The majority of included studies were retrospective observational studies [21,22,23,46,47,48,50,51,53,54,55,56,57,58,60,61,62,63,64,65,66,68,69,70,71,72,73,74,75,76,77,78,79,80,81,82,83,84,85,86,87,88,89,90,91,92,93] rather than randomized controlled trials, lowering the overall general quality of evidence. Furthermore, several secondary outcomes were quantified, analyzed and reported but not discussed in detail (Appendix A). The decision was taken to do so, as the main aim of this study was to discuss relevant postoperative complications especially important for surgical practice. 

Furthermore, the influence of preoperative timing of anti-TNF-α biologic drug administration on postoperative complications could not be statistically evaluated as the included studies only depict a preoperative time frame, rather than a specific time point at which these biologics were administered [21,22,23,46,47,48,49,50,51,52,53,54,55,56,57,58,59,60,61,62,63,64,65,66,67,68,69,70,71,72,73,74,75,76,77,78,79,80,81,82,83,84,85,86,87,88,89,90,91,92,93].

Finally, potential confounding factors (such as disease severity, comorbidities, concomitant immunosuppression, to name a few) may have influenced or even biased the outcomes of the analyses. This important limitation was addressed by conducting extensive subgroup analysis according to the most commonly reported risk factors. Furthermore, in case of significant or substantial heterogeneity sensitivity analysis was performed to evaluate the effect of excluding one study at a time, on pooled OR. 

However, the strength of this study is that it presents the first systematic review and meta-analysis investigating solely patients that have been treated with anti-TNF-α biologics within the washout period of 12 weeks [18] (or detectable serum concentrations [21]) prior to surgery and were followed up exactly 30 days postoperatively. Furthermore, to our knowledge, such extensive subgroup analyses stratified for both the IBD subtype and potential risk factors, were not performed before. [3,4,5,6,7,8,9,10,11,12,13,14,15,16,17].

In addition, we eliminated many sources of bias, still found in other meta-analysis, e.g., (1)The overall postoperative complication rate was calculated individually, summation of all individual complications was not performed.(2)The correct number of patients was included for all studies evaluated in this analysis.(3)The study offers an insight on how authors assessed for risk of bias in accordance with the NOS scale—authors’ judgment and rationale, underlined by citations, were presented (Appendix A).

The outcomes of this systematic review and meta-analysis present some clinical implications, not at least due to the above-mentioned strengths and transparent working processes. The outcomes of this study suggest that the treatment with anti-TNF-α biologic drugs within the washout period of 12 weeks [18] (or detectable serum concentrations [21]) prior to abdominal surgery are associated with important short-term postoperative general and surgical complications for patients with IBD, in particular CD. Still, future prospective studies need to be conducted to assess the magnitude of the influence of preoperative timing of anti-TNF-α treatment and potential confounding factors on postoperative outcomes. 

## 5. Conclusions

In conclusion, current evidence suggests that the treatment with anti-TNF-α agents within the washout period of 12 weeks (3 months [18] or detectable serum concentrations [21]) prior to abdominal surgery is associated with important short-term general and surgical postoperative complications for patients with IBD, in particular with CD. With regard to postoperative mortality, on the one hand, the concomitant treatment with corticosteroids and/or immunomodulatory drugs could be detrimental for patients preoperatively treated with anti-TNF-α biologics, which is why those patients might benefit from a reduction of perioperative immunosuppression, if possible. On the other hand, perioperative immunosuppressive therapy could have a protective effect for IBD patients not treated with anti-TNF-α preoperatively.

## Figures and Tables

**Figure 1 jcm-11-06884-f001:**
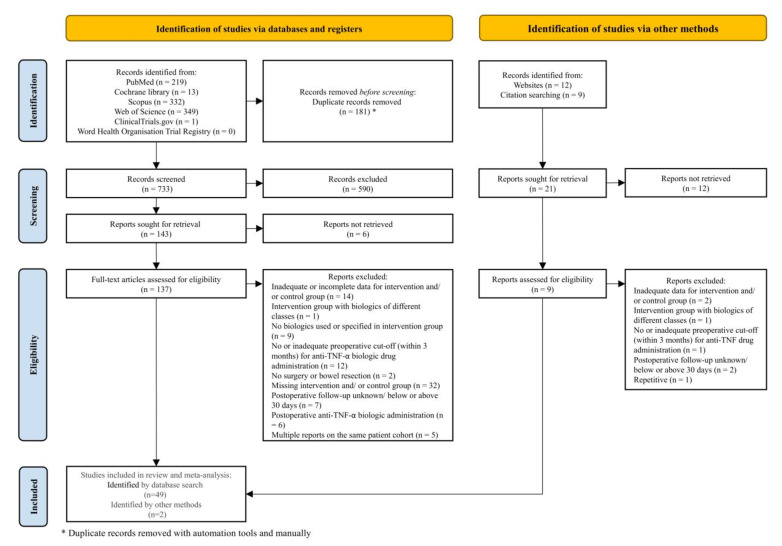
The study flow diagram according to the Preferred Reporting Items for Systematic Review and Meta-Analyses (PRISMA) Statement 2020 [20].

**Figure 2 jcm-11-06884-f002:**
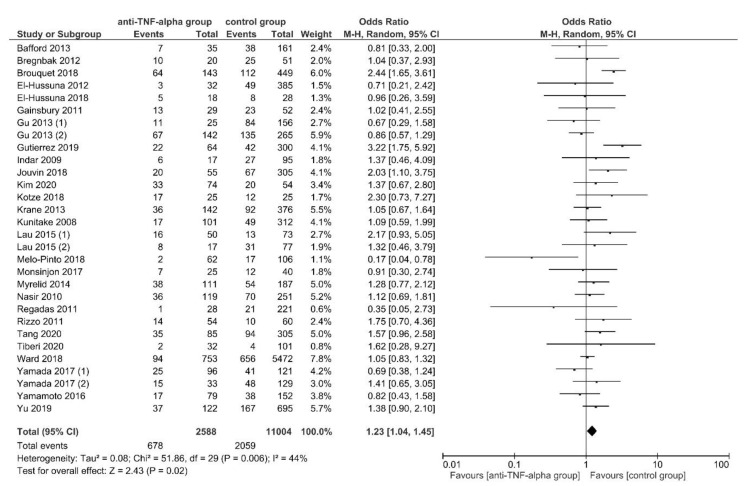
Random-effects model meta-analysis for 30-day overall postoperative complications in the anti-TNF-α (intervention) and control group. Forest plot of all studies included [21,22,23,47,48,49,51,52,54,55,56,57,58,59,61,62,63,67,69,70,74,75,81,86,89,90,91].

**Figure 3 jcm-11-06884-f003:**
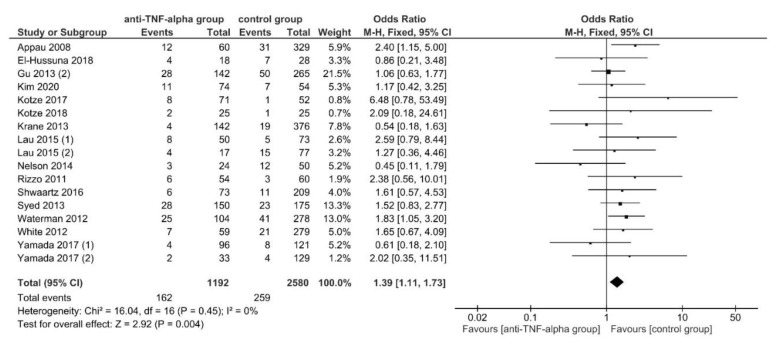
The fixed-effects model meta-analysis for 30-day postoperative readmission in the anti-TNF-α (intervention) and control group. Forest plot of all studies included. [21,46,52,55,59,60,61,62,71,75,78,79,87,88,89].

**Figure 4 jcm-11-06884-f004:**
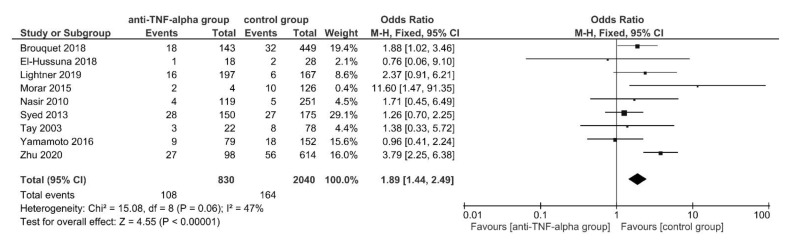
The fixed-effects model meta-analysis for 30-day postoperative intra-abdominal septic complications in the anti-TNF-α (intervention) and control group. Forest plot of all studies included. [49,52,64,68,70,79,80,90,92].

**Table 1 jcm-11-06884-t001:** Study and patient characteristics.

Author and Year	Country	Years of Study	IBD Subtype ^†^	Type of Anti-TNF-α Agent ^‡^	Last Preoperative Anti-TNF-α Agent Exposure in Weeks	Duration of Postoperative Follow-Up in Days	Number of Patients		Number of Patients with Concomitant Preoperative Medication Use: Corticosteroids/Immunomodulators		NOS Score ^¶^
				I ^§^	I ^§^	I ^§^/C ^§^	I ^§^	C ^§^	I ^§^	C ^§^	
Appau et al., 2008 [46]	USA	1998–2007	CD	IFX	<12	30	60	329	39/n.a.	253/n.a.	7
Bafford et al., 2013 [47]	USA	1999–2010	CD	U	<12	30	35	161	16/19	56/50	5
Bregnbak et al., 2012 [48]	Denmark	2005–2010	UC	IFX	<12	30	20	51	15/n.a.	33/n.a.	5
Brouquet et al., 2018 [49]	France	2013–2015	CD	ADA, IFX, U	<12	30	143	449	1/n.a.	46/n.a.	7
Canedo et al., 2010 [50]	USA	2001–2008	CD	IFX	<12	30	65	160	n.a.	n.a.	6
El-Hussuna et al., 2012 [51]	Denmark	2000–2007	CD	CZP, IFX	<12	30	32	385	28/11	135/135	6
El-Hussuna et al., 2018 [52]	Denmark	2014–2016	IBD (CD/UC)	ADA, GO, IFX, U	<12	30	18	28	7/8	9/7	6
Ferrante et al., 2009 [53]	Belgium	1998–2008	UC and IC	IFX	<12	30	22	119	14/13	82/65	5
Gainsbury et al., 2011 [54]	USA	2005–2009	UC	IFX	<12	30	29	52	27/n.a.	36/n.a.	7
Gu et al., 2013 *(1)* * [55]	USA	2006–2010	UC and IC	ADA, IFX,	4–12	30	25	156	10/7	80/41	5
Gu et al., 2013 *(2)* * [55]				ADA, CZP, IFX			142	265	100/34	212/74	
Gutierrez et al., 2019 [56]	Spain	2007–2010	CD	U	0	30	64	300	n.a.	n.a.	7
Indar et al., 2009 [57]	USA	1999–2007	CD	IFX	<6	30	17	95	10/8	37/31	5
Jouvin et al., 2018 [58]	France	2002–2013	CD	ADA, CZP, IFX	<8	30	55	305	15/29	139/118	4
Kim et al., 2020 [59]	USA	2007–2017	UC	ADA, IFX	<12	30	74	54	48/62	34/24	7
Kotze et al., 2017 [61]	Brazil	2007–2014	CD	ADA, IFX	<8	30	71	52	22/58	20/33	7
Kotze et al., 2018 [60]	Brazil	7-year period	CD	ADA	<8	30	25	25	8/22	11/17	6
Krane et al., 2013 [62]	USA	2004–2011	IBD (CD/IC/UC)	IFX	<12	30	142	376	105/46	221/77	8
Kunitake et al., 2008 [63]	USA	1993–2007	IBD (CD/IC/UC)	IFX	<12	30	101	312	76/37	240/81	5
Lau et al., 2015 *(1)* * [21]	USA	1999–2012	CD	ADA, CZP, IFX	0	30	50	73	43/n.a.	56/n.a.	6
Lau et al., 2015 *(2)* * [21]			UC			30	17	77	4/n.a.	13/n.a.	
Lightner et al., 2019 [64]	USA	2014–2017	CD	ADA, CZP, IFX	<12	30	197	105	37/18	22/38	6
Maeda et al., 2015 [65]	Japan	2005–2013	CD	ADA, IFX	<4	30	65	112	n.a.	n.a.	4
Mascarenhas et al., 2012 [66]	USA	2003–2010	CD	IFX	<12	30	19	74	n.a.	n.a.	5
Melo-Pinto et al., 2018 [22]	Portugal	2010–2015	CD	U	<12	30	62	106	13/n.a.	33/n.a.	7
Monsinjon et al., 2017 [67]	France	2006–2015	IBD (CD/UC)	U	0	30	25	40	n.a.	n.a.	5
Morar et al., 2015 [68]	UK	2005–2010	CD	ADA, IFX	<4	30	4	126	n.a.	n.a.	6
Myrelid et al., 2014 [69]	Sweden	2005–2011	CD	ADA, IFX	<8	30	111	187	n.a.	n.a.	5
Nasir et al., 2010 [70]	USA	2005–2009	CD	ADA, CZP, IFX	<8	30	119	251	16/63	25/110	5
Nelson et al., 2014 [71]	USA	2006–2012	UC	IFX	<4	30	24	50	37/32	114/83	5
Nørgård et al., 2012 [72]	Denmark	2003–2010	UC	ADA, IFX, U	<12	30	199	1027 ^(D/M)^/997 ^(C)^	24/0	50/19	6
Nørgård et al., 2013 [73]	Denmark	2000–2010	CD	ADA, IFX, U	<12	30	214 ^(D/M)^/213 ^(C)^	2079 ^(D/M)^/2025 ^(C)^	49/n.a.	300/n.a.	6
Regadas et al., 2011 [74]	USA	2001–2008	IBD (CD/IC/UC)	IFX	<8	30	28	221	19/n.a.	294/n.a.	5
Rizzo et al., 2011 [75]	Italy	2004–2010	IBD (CD/UC)	ADA, CZP, IFX	<12	30	54	60	18/15	107/35	7
Selvasekar et al., 2007 [76]	USA	2002–2005	UC	IFX	<8	30	23	254	19/21	29/6	7
Serradori et al., 2013 [77]	France	2000–2010	CD	ADA, IFX	<12	30	42	175	n.a.	n.a.	5
Shwaartz et al., 2016 [78]	USA	2013–2015	IBD (CD/UC)	ADA, CZP, IFX	<8	30	73	209	16/n.a.	71/n.a.	6
Syed et al., 2013 [79]	USA	2004–2011	CD	ADA, CZP, IFX	<8	30	150	175	26/29	50/55	5
Tang et al., 2020 [23]	China	2014–2018	CD	IFX	<8	30	85	305	23/53	30/63	6
Tay et al., 2003 [80]	USA	1998–2002	CD	IFX	<8	30	22	78	35/n.a.	85/n.a.	7
Tiberi et al., 2020 [81]	Japan	2008–2019	CD	U	<12	30	32	101	0/n.a.	14/n.a.	5
Uchino et al., 2013 [84]	Japan	2008–2011	CD	IFX	<12	30	79	326	n.a.	n.a.	6
Uchino et al., 2013 [85]	Japan	2010–2012	UC	IFX	<12	30	22	174	29/1	110/5	7
Uchino et al., 2015 [83]	Japan	2012–2014	UC	U	<12	30	44	137	17/14	104/80	7
Uchino et al., 2019 [82]	UK	2015–2018	UC	ADA, GO, IFX	<12	30	146	155	n.a.	n.a.	7
Ward et al., 2018 [86]	Canada	2006–2015	UC	ADA, GO, IFX	<12	30	753	5472	n.a.	n.a.	6
Waterman et al., 2012 [87]	USA	2000–2010	IBD (CD/IC/UC)	AD, IFX	<12	30	104	278	n.a.	n.a.	8
White et al., 2012 [88]	USA	1999–2009	CD	ADA, CZP, IFX	<12	30	59	279	24/35	47/56	5
Yamada et al., 2017 *(1)* * [89]	USA	2014–2016	CD	ADA, IFX, U	<4	30	96	121	n.a.	n.a.	8
Yamada et al., 2017 *(2)* * [89]			UC				33	129			
Yamamoto et al., 2016 [90]	Japan	2008–2013	CD	ADA, IFX	<8	30	79	152	n.a.	n.a.	8
Yu et al., 2019 [91]	Korea	2006–2015	CD	ADA, IFX	<8	30	122	339	26/31	42/34	5
Zhu et al., 2020 [92]	China	2011–2017	CD	IFX	<8	30	98	614	20/66	85/296	6
Zittan et al., 2016 [93]	Canada	2002–2013	UC	U	<4	30	27	562	n.a.	n.a.	7

n.a. = Not available, (D/M) = Demographics/mortality, (C) = Complications, * Gu et al., 2013, Lau et al. and 2015, Yamada et al., 2017 include two discrete data sets each, ^†^ CD = Crohn’s disease, IBD = Inflammatory bowel disease, IC = Indeterminate colitis, UC = Ulcerative colitis, ^‡^ ADA = Adalimumab, CZP = Certolizumab pegol, GO = Golimumab, IFX = Infliximab, TNF = Tumor necrosis factor, U = Unspecific (any anti-TNF-α biologic), ^§^ I = Intervention group (anti-TNF-α biologic exposure within 12 weeks prior to intestinal surgery), C = Control group, ^¶^ NOS = Newcastle-Ottawa Scale (out of 9 stars).

**Table 2 jcm-11-06884-t002:** 30-day general and surgical-site postoperative complications.

Author and Year	General Postoperative Complications ^†^												Surgical-Site Postoperative Complications ^‡^							
	OPC, n		OIC, n		OC-DMC, n		RA, n		RO, n		M, n		OISSC, n		AL, n		IAA, n		IASC, n	
	I ^§^	C ^§^	I ^§^	C ^§^	I ^§^	C ^§^	I ^§^	C ^§^	I ^§^	C ^§^	I ^§^	C ^§^	I ^§^	C ^§^	I ^§^	C ^§^	I ^§^	C ^§^	I ^§^	C ^§^
Appau et al., 2008 [46]	n.a.	n.a.	n.a.	n.a.	n.a.	n.a.	**↑12**	**↓31**	5	10	1	0	n.a.	n.a.	6	14	6	14	n.a.	n.a.
Bafford et al., 2013 [47]	7	38	n.a.	n.a.	n.a.	n.a.	n.a.	n.a.	n.a.	n.a.	0	0	n.a.	n.a.	n.a.	n.a.	n.a.	n.a.	n.a.	n.a.
Bregnbak et al., 2012 [48]	10	25	4	21	n.a.	n.a.	n.a.	n.a.	n.a.	n.a.	0	0	n.a.	n.a.	n.a.	n.a.	1	2	n.a.	n.a.
Brouquet et al., 2018 [49]	**↑64**	**↓112**	n.a.	n.a.	n.a.	n.a.	n.a.	n.a.	n.a.	n.a.	0	0	n.a.	n.a.	n.a.	n.a.	n.a.	n.a.	**↑18**	**↓32**
Canedo et al., 2010 [50]	n.a.	n.a.	n.a.	n.a.	n.a.	n.a.	n.a.	n.a.	2	9	n.a.	n.a.	n.a.	n.a.	2	5	2	8	n.a.	n.a.
El-Hussuna et al., 2012 [51]	3	49	n.a.	n.a.	n.a.	n.a.	n.a.	n.a.	n.a.	n.a.	n.a.	n.a.	n.a.	n.a.	n.a.	n.a.	n.a.	n.a.	n.a.	n.a.
El-Hussuna et al., 2018 [52]	5	8	n.a.	n.a.	n.a.	n.a.	4	7	n.a.	n.a.	n.a.	n.a.	n.a.	n.a.	n.a.	n.a.	n.a.	n.a.	1	2
Ferrante et al., 2009 [53]	n.a.	n.a.	2	29	n.a.	n.a.	n.a.	n.a.	n.a.	n.a.	0	0	1	23	0	15	0	9	n.a.	n.a.
Gainsbury et al., 2011 [54]	13	23	5	14	n.a.	n.a.	n.a.	n.a.	n.a.	n.a.	0	0	n.a.	n.a.	n.a.	n.a.	4	7	n.a.	n.a.
Gu et al., 2013 *(1)* * [55]	11	84	n.a.	n.a.	n.a.	n.a.	n.a.	n.a.	n.a.	n.a.	n.a.	n.a.	n.a.	n.a.	n.a.	n.a.	n.a.	n.a.	n.a.	n.a.
Gu et al., 2013 *(2)* * [55]	67	135	n.a.	n.a.	n.a.	n.a.	28	50	8	20	2	1	n.a.	n.a.	16	24	3	12	n.a.	n.a.
Gutierrez et al., 2019 [56]	22	42	n.a.	n.a.	n.a.	n.a.	n.a.	n.a.	n.a.	n.a.	n.a.	n.a.	n.a.	n.a.	n.a.	n.a.	n.a.	n.a.	n.a.	n.a.
Indar et al., 2009 [57]	6	27	n.a.	n.a.	n.a.	n.a.	n.a.	n.a.	n.a.	n.a.	0	0	n.a.	n.a.	n.a.	n.a.	n.a.	n.a.	n.a.	n.a.
Jouvin et al., 2018 [58]	**↑20**	**↓67**	**↑14**	**↓44**	n.a.	n.a.	n.a.	n.a.	n.a.	n.a.	1	2	n.a.	n.a.	n.a.	n.a.	n.a.	n.a.	n.a.	n.a.
Kim et al., 2020 [59]	33	20	8	11	n.a.	n.a.	11	7	0	0	n.a.	n.a.	n.a.	n.a.	n.a.	n.a.	2	0	n.a.	n.a.
Kotze et al., 2017 [61]	n.a.	n.a.	n.a.	n.a.	n.a.	n.a.	8	1	12	7	2	2	26	11	6	6	10	2	n.a.	n.a.
Kotze et al., 2018 [60]	17	12	n.a.	n.a.	7	9	2	1	4	4	0	1	9	6	2	4	3	2	n.a.	n.a.
Krane et al., 2013 [62]	36	92	17	42	12	32	4	19	5	9	0	1	n.a.	n.a.	3	5	8	20	n.a.	n.a.
Kunitake et al., 2008 [63]	17	49	6	32	n.a.	n.a.	n.a.	n.a.	n.a.	n.a.	2	1	n.a.	n.a.	3	9	n.a.	n.a.	n.a.	n.a.
Lau et al., 2015 *(1)* * [21]	16	13	11	7	n.a.	n.a.	8	5	n.a.	n.a.	1	0	n.a.	n.a.	n.a.	n.a.	n.a.	n.a.	n.a.	n.a.
Lau et al., 2015 *(2)* * [21]	8	31	2	10	n.a.	n.a.	4	15	n.a.	n.a.	0	0	n.a.	n.a.	n.a.	n.a.	n.a.	n.a.	n.a.	n.a.
Lightner et al., 2019 [64]	n.a.	n.a.	33	30	n.a.	n.a.	n.a.	n.a.	n.a.	n.a.	n.a.	n.a.	n.a.	n.a.	n.a.	n.a.	n.a.	n.a.	16	6
Maeda et al., 2015 [65]	n.a.	n.a.	n.a.	n.a.	n.a.	n.a.	n.a.	n.a.	n.a.	n.a.	n.a.	n.a.	8	27	n.a.	n.a.	n.a.	n.a.	n.a.	n.a.
Mascarenhas et al., 2012 [66]	n.a.	n.a.	n.a.	n.a.	2	3	n.a.	n.a.	n.a.	n.a.	0	0	n.a.	n.a.	n.a.	n.a.	n.a.	n.a.	n.a.	n.a.
Melo-Pinto et al., 2018 [22]	2	17	n.a.	n.a.	n.a.	n.a.	n.a.	n.a.	n.a.	n.a.	n.a.	n.a.	n.a.	n.a.	n.a.	n.a.	n.a.	n.a.	n.a.	n.a.
Monsinjon et al., 2017 [67]	7	12	n.a.	n.a.	n.a.	n.a.	n.a.	n.a.	n.a.	n.a.	0	0	n.a.	n.a.	n.a.	n.a.	n.a.	n.a.	n.a.	n.a.
Morar et al., 2015 [68]	n.a.	n.a.	n.a.	n.a.	n.a.	n.a.	n.a.	n.a.	n.a.	n.a.	n.a.	n.a.	n.a.	n.a.	n.a.	n.a.	n.a.	n.a.	**↑2**	**↓10**
Myrelid et al., 2014 [69]	38	54	18	26	12	20	n.a.	n.a.	9	13	n.a.	n.a.	n.a.	n.a.	8	15	n.a.	n.a.	n.a.	n.a.
Nasir et al., 2010 [70]	36	70	n.a.	n.a.	n.a.	n.a.	n.a.	n.a.	n.a.	n.a.	0	1	n.a.	n.a.	n.a.	n.a.	n.a.	n.a.	4	5
Nelson et al., 2014 [71]	n.a.	n.a.	6	12	n.a.	n.a.	3	12	n.a.	n.a.	0	0	4	8	n.a.	n.a.	2	2	n.a.	n.a.
Nørgård et al., 2012 [72]	n.a.	n.a.	n.a.	n.a.	n.a.	n.a.	n.a.	n.a.	43	230	0	30	n.a.	n.a.	1	16	n.a.	n.a.	n.a.	n.a.
Nørgård et al., 2013 [73]	n.a.	n.a.	n.a.	n.a.	n.a.	n.a.	n.a.	n.a.	16	175	1	54	n.a.	n.a.	8	56	n.a.	n.a.	n.a.	n.a.
Regadas et al., 2011 [74]	1	21	0	22	n.a.	n.a.	n.a.	n.a.	0	9	0	0	n.a.	n.a.	0	3	0	5	n.a.	n.a.
Rizzo et al., 2011 [75]	14	10	9	8	n.a.	n.a.	6	3	3	1	0	0	n.a.	n.a.	4	3	n.a.	n.a.	n.a.	n.a.
Selvasekar et al., 2007 [76]	n.a.	n.a.	6	25	n.a.	n.a.	n.a.	n.a.	n.a.	n.a.	0	0	n.a.	n.a.	n.a.	n.a.	n.a.	n.a.	n.a.	n.a.
Serradori et al., 2013 [77]	n.a.	n.a.	n.a.	n.a.	n.a.	n.a.	n.a.	n.a.	n.a.	n.a.	0	0	9	15	n.a.	n.a.	n.a.	n.a.	n.a.	n.a.
Shwaartz et al., 2016 [78]	n.a.	n.a.	n.a.	n.a.	n.a.	n.a.	6	11	n.a.	n.a.	0	0	n.a.	n.a.	4	11	4	4	n.a.	n.a.
Syed et al., 2013 [79]	n.a.	n.a.	**↑54**	**↓44**	47	47	28	23	24	23	2	1	n.a.	n.a.	9	9	21	18	28	27
Tang et al., 2020 [23]	35	94	**↑22**	**↓43**	19	61	n.a.	n.a.	n.a.	n.a.	n.a.	n.a.	24	50	5	18	2	6	n.a.	n.a.
Tay et al., 2003 [80]	n.a.	n.a.	3	8	n.a.	n.a.	n.a.	n.a.	n.a.	n.a.	n.a.	n.a.	n.a.	n.a.	1	3	2	4	3	8
Tiberi et al., 2020 [81]	2	4	n.a.	n.a.	2	4	n.a.	n.a.	n.a.	n.a.	0	0	n.a.	n.a.	n.a.	n.a.	n.a.	n.a.	n.a.	n.a.
Uchino et al., 2013 [84]	n.a.	n.a.	n.a.	n.a.	n.a.	n.a.	n.a.	n.a.	n.a.	n.a.	n.a.	n.a.	**↓9**	**↑99**	n.a.	n.a.	n.a.	n.a.	n.a.	n.a.
Uchino et al., 2013 [85]	n.a.	n.a.	n.a.	n.a.	n.a.	n.a.	n.a.	n.a.	n.a.	n.a.	n.a.	n.a.	**↓1**	**↑46**	n.a.	n.a.	n.a.	n.a.	n.a.	n.a.
Uchino et al., 2015 [83]	n.a.	n.a.	n.a.	n.a.	n.a.	n.a.	n.a.	n.a.	n.a.	n.a.	n.a.	n.a.	5	32	n.a.	n.a.	n.a.	n.a.	n.a.	n.a.
Uchino et al., 2019 [82]	n.a.	n.a.	n.a.	n.a.	n.a.	n.a.	n.a.	n.a.	n.a.	n.a.	n.a.	n.a.	16	25	n.a.	n.a.	n.a.	n.a.	n.a.	n.a.
Ward et al., 2018 [86]	94	656	35	270	n.a.	n.a.	n.a.	n.a.	n.a.	n.a.	n.a.	n.a.	n.a.	n.a.	n.a.	n.a.	n.a.	n.a.	n.a.	n.a.
Waterman et al., 2012 [87]	n.a.	n.a.	n.a.	n.a.	n.a.	n.a.	25	41	3	12	1	1	n.a.	n.a.	2	15	n.a.	n.a.	n.a.	n.a.
White et al., 2012 [88]	n.a.	n.a.	n.a.	n.a.	n.a.	n.a.	7	21	n.a.	n.a.	n.a.	n.a.	n.a.	n.a.	n.a.	n.a.	n.a.	n.a.	n.a.	n.a.
Yamada et al., 2017 *(1)* * [89]	25	41	10	16	n.a.	n.a.	4	8	6	4	0	0	n.a.	n.a.	2	2	6	8	n.a.	n.a.
Yamada et al., 2017 *(2)* * [89]	15	48	9	18	n.a.	n.a.	2	4	2	8	0	0	n.a.	n.a.	2	2	2	10	n.a.	n.a.
Yamamoto et al., 2016 [90]	17	38	n.a.	n.a.	n.a.	n.a.	n.a.	n.a.	n.a.	n.a.	n.a.	n.a.	n.a.	n.a.	7	12	n.a.	n.a.	9	18
Yu et al., 2019 [91]	37	167	28	126	n.a.	n.a.	n.a.	n.a.	n.a.	n.a.	n.a.	n.a.	n.a.	n.a.	n.a.	n.a.	n.a.	n.a.	n.a.	n.a.
Zhu et al., 2020 [92]	n.a.	n.a.	n.a.	n.a.	n.a.	n.a.	n.a.	n.a.	n.a.	n.a.	n.a.	n.a.	n.a.	n.a.	n.a.	n.a.	n.a.	n.a.	**↑27**	**↓56**
Zittan et al., 2016 [93]	n.a.	n.a.	n.a.	n.a.	n.a.	n.a.	n.a.	n.a.	n.a.	n.a.	n.a.	n.a.	n.a.	n.a.	2	66	3	96	n.a.	n.a.

n = Number, n.a. = Not available, Bold marked = Significant outcome (↓ = significantly lower, ↑ = significantly higher), * Gu et al., 2013, Lau et al. and 2015, Yamada et al., 2017 include two discrete data sets each, ^†^ OPC = Overall postoperative complications, OIC = Overall infectious postoperative complications, OC-DMC = Overall Clavien-Dindo major complications, RA = Readmission rate, RO = Reoperation rate, M = Mortality rate, ^‡^ OISSC = Overall infectious surgical-site complications, AL = Anastomotic leakage, IAA = Intra-abdominal abscess, IASC = Intra-abdominal septic complications, ^§^ I = Intervention group (anti-TNF-α drug exposure within 12 weeks prior to intestinal surgery), C = Control group.

**Table 3 jcm-11-06884-t003:** The fixed-effect model meta-analysis for 30-day postoperative readmission and intra-abdominal abscesses in the anti-TNF-α (intervention) and control group. Analysis for publication bias and subgroup analysis.

Readmission							Intra-Abdominal Abscesses						
OR with 95% CI–FE			Heterogeneity		Egger’s Test [45]		OR with 95% CI–FE			Heterogeneity		Egger’s Test [45]	
OR, 1.39; 95% CI, 1.11–1.73; ***p* = 0.004 [↑ (I); ↓(C)]**			*I^2^* = 0%; *p* = 0.45		*p* = 0.994		OR, 1.19; 95% CI, 0.89–1.60; *p* = 0.24			*I^2^* = 0%; *p* = 0.76		*p* = 0.850	
**Subgroup analysis**													
**Subgroups**		**OR with 95% CI–FE**		**TSD**	**SA**	**TSD**	**Subgroups**		**OR with 95% CI–FE**		**TSD**	**SA**	**TSD**
**IBD**	CD	OR, 1.77; 95% CI, 1.23–2.52; ***p* = 0.002 [↑ (I); ↓(C)]**		*p* = 0.15	CD	** *p* ** ** = 0.05**	**IBD**	CD	OR, 1.50; 95% CI, 1.00–2.25; ***p* = 0.05 [↑ (I); ↓(C)]**		*p* = 0.15	CD	** *p* ** ** = 0.05**
	UC/IC	OR, 1.04; 95% CI, 0.70–1.54; *p* = 0.86			UC/IC			UC/IC	OR, 0.76; 95% CI, 0.43–1.32; *p* = 0.33			UC/IC	
	IBD	OR, 1.40; 95% CI, 0.94–2.08; *p* = 0.10			x			IBD	OR, 1.30; 95% CI, 0.66–1.60; *p* = 0.45			x	
**Potential risk factors**													
**OA**	>50%	OR, 2.01; 95% CI, 1.32–3.08; ***p* = 0.001 [↑ (I); ↓(C)]**		*p* = 0.06	>50%	** *p* ** ** = 0.02**	**OA**	>50%	OR, 1.37; 95% CI, 0.91–2.06; *p* = 0.13		*p* = 0.62	>50%	*p* = 0.35
	<50%	OR, 1.01; 95% CI, 0.70–1.47; *p* = 0.94			<50%			<50%	OR, 1.00; 95% CI, 0.58–1.70; *p* = 0.99			<50%	
	U	OR, 1.43; 95% CI, 1.00–2.07; ***p =* 0.05 [↑ (I); ↓(C)]**			x			U	OR, 1.09; 95% CI, 0.55–2.16; *p* = 0.80			x	
**ELEMS**	EL	OR, 1.21; 95% CI, 0.69–2.13; *p* = 0.50		*p* = 0.35	EL	*p* = 0.98	**ELEMS**	EL	OR, 1.65; 95% CI, 0.95–2.87; *p* = 0.07		*p* = 0.38	EL	*p* = 0.16
	ELEMS	OR, 1.22; 95% CI, 0.88–1.70; *p* = 0.23			ELEMS			ELEMS	OR, 1.01; 95% CI, 0.66–1.54; *p* = 0.96			ELEMS	
	U	OR, 1.70; 95% CI, 1.20–2.40; ***p* = 0.003 [↑ (I); ↓(C)]**			x			U	OR, 1.16; 95% CI, 0.63–2.12; *p* = 0.64			x	
**PRI**	<50%	OR, 1.79; 95% CI, 1.32–2.42; ***p* = 0.0002 [↑ (I); ↓(C)]**		*p* = 0.06	<50%	** *p* ** ** = 0.03**	**PRI**	<50%	OR, 1.58; 95% CI, 1.06–2.37; ***p* = 0.02 [↑ (I); ↓(C)]**		*p* = 0.13	<50%	** *p* ** ** = 0.05**
	>50%	OR, 1.07; 95% CI, 0.76–1.51; *p* = 0.69			>50%			>50%	OR, 0.80; 95% CI, 0.47–1.37; *p* = 0.42			>50%	
	U	OR, 0.87; 95% CI, 0.32–2.38; *p* = 0.79			x			U	OR, 1.03; 95% CI, 0.48–2.20; *p* = 0.94			x	
**CSIM**	>50%	OR, 1.27; 95% CI, 0.92–1.74; *p* = 0.15		*p* = 0.73	>50%	*p* = 0.56	**CSIM**	>50%	OR, 1.28; 95% CI, 0.86–1.91; *p* = 0.22		*p* = 0.53	>50%	*p* = 1.0
	<50%	OR, 1.48; 95% CI, 0.96–2.28; *p* = 0.07			<50%			<50%	OR, 1.28; 95% CI, 0.76–2.16; *p* = 0.35			<50%	
	U	OR, 1.54; 95% CI, 1.01–2.37; *p* = 0.05			x			U	OR, 0.80; 95% CI, 0.37–1.72; *p* = 0.56			x	

Bold marked = Significant outcome (↓ = significantly lower; ↑ = significantly higher), I = Intervention group (anti-TNF-α drug exposure within 12 weeks prior to intestinal surgery), C = Control group, OR = Odds ratio, CI = Confidence interval, FE = Fixed-effects model meta-analysis, TSD = Test for subgroup difference, IBD = Inflammatory bowel disease, CD = Crohn’s disease, UC = Ulcerative colitis, IC = Indeterminate colitis, OA = Open surgery or conversion to open surgery, U = Unknown, EL = Elective surgery, ELEMS = Elective surgery and emergency surgery, PRI = Protective ileostomy, CSIM = Concomitant corticosteroid and/or immunomodulatory drug administration.

## Data Availability

The data used to support the findings of this systematic review and meta-analysis are included within the article.

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
