# Peer review of "The Effect of Anti-Tumor Necrosis Factor-Alpha Therapy within 12 Weeks Prior to Surgery on Postoperative Complications in Inflammatory Bowel Disease: A Systematic Review and Meta-Analysis"

_jcm, 2022, doi:10.3390/jcm11236884_

Round 1

Reviewer 1 Report

The authors make efforts to evaluate the effect of anti-TNF-α treatment exclusively within the washout period of 12 weeks (3 months) prior to abdominal surgery, on 30-day general and surgical postoperative complication rate in IBD patients. The author searched a large number of documents and conclude that “Anti-TNF-α treatment within 12 weeks prior to surgery is associated with higher short-term postoperative complication rates”. However, I have some comments on this study.

1.      The vast majority of IBD patients may be treated with these drugs instead of surgery, including anti-TNF-α treatment. Then what is the clinical significance of this study? The authors should give some explanation in the discussion.

2.      The discussion is too superficial, just a repetition of the result. It should be better to give a more detailed illustration.

3.      In the discussion, you had better to suppose that why the short-term postoperative complication rates is higher when treated with anti-TNF-α treatment within 12 weeks prior to surgery?

4.      The patients with IBD may be treated with anti-TNF-α over a long period of time, Why you choose “12 weeks prior to surgery” as your objects?

5.      The short-term postoperative complication is maybe affected by the operation itself, then how much does the drug anti-TNF-α affect these complications?

Author Response

Dear Reviewer,

we highly appreciate your feedback.

Also we have formulated a point-by-point response to your comments (see below). 

We would like to thank you again and hope that our manuscript now meets your expectations.

Kind regards

Kamacay Cira

Point to point response to Reviewer’s comments

Reviewer 1

Question 1:  The vast majority of IBD patients may be treated with these drugs instead of surgery, including anti-TNF-α treatment. Then what is the clinical significance of this study? The authors should give some explanation in the discussion.

Answer: Thank you very much for this comment. Today, up to 40% of CD patients and up to 16% of UC patients are under treatment with anti-TNF-α biologics such as Adalimumab, Infliximab, Golimumab, Certolizumab (pegol) or one of its biosimilars. However, in comparison to the general population, the 5-year surgery rates with its accompanying potential postoperative complications remain substantial for IBD patients.

To reduce the total clinical and economic burden for IBD patients undergoing abdominal surgery it is of great importance to improve the perioperative treatment of IBD patients and to address every possible risk factor for postoperative complications.

To serve as an orientation for surgeons and provide an implication into surgical practice, this review and the resulting meta-analysis primarily focused on 30-day postoperative general and surgical complication rates. The outcomes of this study suggest that the treatment with anti-TNF-α biologic drugs within the washout period of 12 weeks (or detectable serum concentrations) prior to abdominal surgery are associated with important short-term postoperative general and surgical complications for patients with IBD, in particular CD.

The outcomes of this study may help surgeons to improve the perioperative therapy in such a way that postoperative complications rates can be reduced to a maximum possible.

We outlined the above mentioned clinical significance in the introduction and discussion of our study and have deepened the discussion.

Question 2:  The discussion is too superficial, just a repetition of the result. It should be better to give a more detailed illustration.

Question 3: In the discussion, you had better to suppose that why the short-term postoperative complication rates is higher when treated with anti-TNF-α treatment within 12 weeks prior to surgery?

Answer for Question 2 and 3: Many thanks for this valuable annotation. Our discussion outlines our findings and compares these with findings of other meta-analysis conducted with the same study question. We analysed the possible reasons for these different findings and underlined our findings with clinical findings of other clinical studies. For every important finding we presumed the possible consequence of treatment and suggested a treatment option in the discussion section.

Question 4: The patients with IBD may be treated with anti-TNF-α over a long period of time, Why you choose “12 weeks prior to surgery” as your objects?

Answer: Thank you very much for this comment. Previously conducted meta-analyses, analyzing the effect of preoperative anti-TNF-α treatment on postoperative complications reported conflicting findings. The latter is mostly the result of including different preoperative drug withdrawal periods in the analyses. To find consistent values regarding the effect of preoperative anti-TNF-α treatment on postoperative complications and to ensure better comparison of data, we chose the objectifiable anti-TNF-α drug washout period of 12 weeks as the preoperative cut-off value in this analysis.

Question 5: The short-term postoperative complication is maybe affected by the operation itself, then how much does the drug anti-TNF-α affect these complications?

Answer: Thank you very much for this valuable question. We compare IBD patients treated with anti-TNF-α biologics within 12 weeks of abdominal surgery with IBD patients who underwent abdominal surgery not treated with anti-TNF-α biologics within 12 weeks. The operation itself may affect the short-term complications in both groups similarly as both groups underwent similar operations. However, to be absolutely sure the outcomes are not biased by surgical risk factors, prespecified subgroup analyses were performed for outcomes reported by ≥ 5 studies (as recommended by the Cochrane Handbook for Reviews of Interventions) to evaluate the influence of IBD subtypes and potential risk factors on the 30-day postoperative complications. Potential risk factors for postoperative complications within each study group were defined a priori as: tobacco use in > 50% of patients; open surgical approach and/ or conversion to open surgery in > 50% of patients (OA); elective and emergency surgery (ELEMS) vs. exclusively elective surgery (EL); performance of a temporary protective ileostomy in < 50% of patients or no use at all (PRI); Body-Mass-Index (BMI) of > 25 kg/m2 or < 18,5 kg/m2; dysplasia or malignancy in > 50% of patients; perforating or penetrating disease as indication for surgery in > 50% of patients; concomitant corticosteroid and/ or immunomodulatory agent use in > 50% of patient (CSIM); laboratory values (median or mean C-reactive protein concentration of > 10 mg/L; white blood cell count > 11 x 109 cells/L or < 4 x 109 cells/L; hemoglobin value of < 10 g/dL; albumin value of < 3 g/dL and platelet count of < 150 x 103/ µl). Influences of these surgical risk factors were reported for every postoperative complication.

Furthermore, to ensure the quality of our results, the heterogeneity between studies was analyzed using the statistical I2 test, considering a I2 of ≥ 50 % as substantial heterogeneity. In case of significant or substantial heterogeneity among the included studies, sensitivity analysis was conducted by evaluating the effect of excluding one study at a time on the pooled OR. Furthermore, we assessed for potential publication bias for outcomes reported by ≥ 10 studies by applying the Egger’s test for funnel plot asymmetry, according to the recommendations of the Cochrane Handbook for Reviews of Interventions.

There are more than 15 previously conducted meta-analyses, analyzing the effect of preoperative anti-TNF-α treatment on postoperative complications after abdominal surgery, presenting conflicting findings mostly due to different methodology of analysis. The strength of this study is that it presents the first systematic review and meta-analysis investigating solely patients that have been treated with anti-TNF-α biologics within the washout period of 12 weeks (or detectable serum concentrations) prior to surgery and were followed up exactly 30 days postoperatively. Furthermore, to our knowledge, such extensive subgroup analyses stratified for both the IBD subtype and potential risk factors, were not performed before.

Reviewer 2 Report

The manuscript by Cira et al. provides a systemic review and meta-analysis of postoperative complications in IBD patients receiving anti-TNF therapy in which they compile studies published between 1995 and 2022. They report that the overall short-term postoperative complications, readmission, and intra-abdominal septic complications are higher for patients receiving anti-TNF treatments. The manuscript is structured, concise, and with clear statements regarding the limitations. As it is an important work for clinical application and is well-written.

Author Response

Dear Reviewer,

we highly appreciate your valuable feedback and would like to thank you that our manuscript meets your expectations.

Kind regards

Kamacay Cira

Round 2

Reviewer 1 Report

The manuscript has been revised and greatly improved. I think it should be  acceptted.